# Assisting Homeless Women in a City in Brazil during the COVID-19 Pandemic in the Context of a Street Outreach Office: The Perceptions of Health Professionals

**DOI:** 10.3390/ijerph20021011

**Published:** 2023-01-05

**Authors:** Nayara Gonçalves Barbosa, Hellen Aparecida de Azevedo Pereira, Marcelo Vinicius Domingos Rodrigues dos Santos, Lise Maria Carvalho Mendes, Flávia Azevedo Gomes-Sponholz, Juliana Cristina dos Santos Monteiro

**Affiliations:** 1Department of Maternal-Child Nursing and Public Health, Faculty of Nursing, Federal University of Juiz de Fora, Juiz de Fora 36036-900, Brazil; 2Public Health Nursing Post-Graduate Program, University of São Paulo at Ribeirão Preto College of Nursing, Ribeirão Preto 14040-902, Brazil; 3Department of Maternal-Child Nursing and Public Health, University of São Paulo at Ribeirão Preto College of Nursing, Ribeirão Preto 14040-902, Brazil

**Keywords:** homeless people, pandemic, COVID-19, gender inequality, women’s health

## Abstract

This study aimed to understand the perception of Street Outreach Office professionals regarding the health care offered to homeless women during the COVID-19 pandemic. This is a qualitative and descriptive study developed with nine health professionals of a Street Outreach Office team from a large city in the countryside of São Paulo State (Brazil) from December 2020 to April 2021. Data were obtained through interviews using a semi-structured script with questions about care practices directed to homeless women. The data were analyzed according to content analysis in the thematic modality. Two thematic categories were identified: (i) the reorganization of the Street Outreach Office to meet the demands of the population and (ii) the challenges in caring for homeless women during the pandemic. The activities were intensified with the team’s expansion and distribution of supplies such as masks and alcohol-based hand sanitizers. Our findings showed that the primary problem faced was pregnancy during the pandemic. The lack of material and structural resources and social apparatus to care for homeless women was also evidenced. It was possible to conclude that even with all the adversities, the professionals employed creative strategies, contributing, within their limitations, to the care of homeless women.

## 1. Introduction

The COVID-19 pandemic, caused by the SARS-CoV-2 virus, had its first cases of infection in late 2019, being declared a pandemic by the World Health Organization (WHO) in March 2020 [1]. In addition to the health consequences, the pandemic brought significant economic, political, and social impacts, which were experienced more intensely by individuals in more vulnerable situations, including homeless people [2,3,4]. Recently, the Technical Note of the Institute for Applied Economic Research (IPEA) reported a 140% increase in the homeless population in Brazil from September 2012 to March 2020 [5]. This population may present a higher risk of infection, transmission, morbidity, and mortality from COVID-19 compared to the general population due to the precarious conditions before the pandemic that worsened the health status of homeless people. These conditions, coupled with the difficulties of access to health and welfare systems, may result in a worse prognosis in cases of infection and recovery from COVID-19 [2,3,6]. Also, thousands of families were exposed to the risk of evictions during the pandemic due the economic crisis and increasing unemployment caused by the restrictive measures. In this sense, there is evidence that moving to crowded shelters or crowding into small houses with family members, to avoid sleeping on the street, were also linked to increasing risk of contracting COVID-19. In this way, it appears that evictions are directly and indirectly associated with the risk of contracting COVID-19 as well [7,8]. It is important to mention that evictions were higher among people of color, low income and women; black poor women had higher rates of eviction when compared to other groups of people [9].

It means, although men appear as a majority in studies on the homeless population, it is highly relevant to consider that, among the consequences of the pandemic faced by all population segments, homeless women require a closer look, since they may not be reliably considered in statistics related to this population [10,11,12]. The higher unemployment rates, lower education levels, and greater chances of suffering violence for women may even be the cause of their homelessness. In addition, this population is more prone to being re-victimized on the streets and in shelters and to higher stress factors and lower satisfaction with their health and empowerment [4,10].

The homeless population faces many difficulties in accessing basic services such as health, social and education services. Often the attendance provided by these services is based on stereotypes that perpetuate prejudices, stigmatizing people who seek care. Added to this are stereotypes and prejudices related to race, gender and social class that create an almost insurmountable barrier to access services and search for their rights. This treatment surrounded by discrimination can make homeless people avoid the search for their rights [13]. These difficulties already evidenced in the scientific literature were exacerbated during the pandemic, with a consequent increase in social inequalities [9,13]. Health services for homeless people have played an essential role in reducing food insecurity, improving hygiene, adherence to vaccination, and, consequently, the protocols for tackling COVID-19 [3]. In Brazil, one of the strategies used for the health care of the homeless population is the Street Outreach Office, a strategy of the National Policy for Primary Care. The Street Outreach Office deals directly with the homeless population’s problems and health needs by carrying out activities in loco, in an itinerant way, and in dialogue with the Basic Health Units (BHU) and other services that comprise the health care network [14]. The Street Outreach Office team consists of a multi-professional team to provide comprehensive and equitable health care and assistance [15].

Considering the vulnerabilities presented by women, which are exacerbated among those living on the streets, and the essential work of Street Outreach Office teams, research contributing to improving and qualifying the performance of these teams is crucial so that they can meet the real demands of these women. Nevertheless, we did not find any studies making this analysis from the perspective of the professionals who work in these teams, especially amid such a turbulent period (i.e., the COVID-19 pandemic), thereby justifying this work.

Given the above, this study sought to shed more light on the perception of health professionals working in the Street Outreach Office regarding the health care provided to homeless women during the COVID-19 pandemic.

## 2. Materials and Methods

### 2.1. Ethical Aspects

The study was approved by the Research Ethics Committee linked to the Brazilian National Research Ethics Committee, in accordance with the principles of Resolution 466/2012 of the Brazilian National Health Council (Conselho Nacional de Saúde). Additionally, the study was authorized by the Municipal Health Department of the studied municipality. All participants signed the informed consent form prior to the beginning of data collection and participated voluntarily in the study. Statements were identified through the letter P, in numerical order of interviews, to protect anonymity.

### 2.2. Study Design and Place

This is a descriptive study, with a qualitative approach, developed in the Street Outreach Office composed of a multidisciplinary team which works on an itinerant basis in several locations in a large city in the countryside of the state of São Paulo, Brazil, from December 2020 to April 2021. The Consolidated Criteria for Reporting Qualitative Research (COREQ) was used to guide the study [16].

### 2.3. Study Participants

All professionals who were active in the Street Outreach Office during the data collection period were invited to participate in the study and all agreed to participate. The participants comprised nine high-school-level and university-level professionals.

### 2.4. Data Collection

As an operational method to carry out the study, data were obtained through semi-structured interviews conducted by only one researcher. The interviews were scheduled with the participants and lasted an average of 45 min; they were held in a private area in the health service center. Safety measures were taken to prevent COVID-19 transmission, including conducting the interview in an airy place, wearing masks, at a minimum distance of one meter and with hand hygiene [17].

The interviews were conducted by the main author of this article, a nurse and professor at a Brazilian higher education institution, with experience in conducting and analyzing qualitative research. The same were recorded in the form of audio, from triggering open questions regarding the care provided to homeless women during the pandemic. Each interview took place in a flexible way, respecting the desire and availability of each participant to talk about the subject and share their experiences. Sociodemographic characteristics of the participants (e.g., age, sex, and education) were collected using a structured form. 

The interviews were audio-recorded and transcribed in full, preserving their originality. In the subsequent stage, floating reading was performed, accompanied by successive readings and a thorough exploration of the material to capture the relevant aspects to answer the research objectives. Thematic content analysis was chosen for data analysis [18,19], according to the phases of pre-analysis and data exploration that were thoroughly carried out and treating the results, inference, and interpretation [18].

The frequency of the themes extracted from the interviews was considered for identifying the central nuclei of meaning, whose presence gives meaning to the proposed objective [19]. Table 1 presents a summary of how the analytical process took place until the construction of two thematic categories, which were: (i) the reorganization of the Street Outreach Office to meet the demands of the population and (ii) the challenges in providing care for homeless women during the COVID-19 pandemic.

## 3. Results

The participants were nine health professionals who worked in the Street Outreach Office: two psychologists, three nurses, one nursing technician, and three community health agents. The average age of the participants was 52 years (standard deviation: 7.3), ranging from 33 to 63 years, and predominantly female, (six, 66.7%). The average time of the workers’ performance was three years and six months.

Two themes were identified in the data analysis: reorganization of the Street Outreach Office to meet the population’s demands; and challenges in the care of homeless women during the COVID-19 pandemic.

### 3.1. Reorganization of the Street Outreach Office to meet the Population’s Demands

The team acts in an itinerant way in public streets, squares, and strategic places known to be frequented by homeless people. In these scenarios, there are meetings between professionals and homeless women.


*(...) We go out to the streets every day, and we visit the squares and the places we have more access to where they stay more on the streets and that we know of (...). We pay a lot of attention to pregnant women we meet at the traffic lights. We try to see this issue of women a lot, especially pregnant women, who do not have access to prenatal care (...). And during the pandemic, we saw a lot of women in these situations.*
P2


*It was more about women’s care, too, giving guidance, delivering supplies, and trying to see if the woman had not been to a health consultation and for how long. We also had stories of women who were pregnant and didn’t go to prenatal care and pregnant women who discovered STIs, including syphilis, after we went to prenatal care.*
P8

The pandemic highlighted the need for expanding the clinic on the Street Outreach Office, considering this population’s demands.


*In the pandemic, as there was an expansion of the team; we were also able to expand the number of people cared for and, consequently, the number of women that we can refer to care (...) we have been working as if it were a mobile BHU and having some resolutions of cases, in addition to referrals.*
P1


*(...) COVID made us work more intensely, made the mobile clinic a reality (...), and women’s care was also more intense at this time. Bringing pregnant women who do not do prenatal care.*
P8

The actions developed specifically during the pandemic included providing supplies, prevention actions, and health orientation related to COVID-19 transmission.


*During the pandemic, besides addressing their issues and needs, which appeared daily, we offered them a mask and alcohol-based hand sanitizers (...); we also took care of ourselves.*
P2


*The mobile clinic was just intensified because of the pandemic, right, to monitor these people in the street, right, at the first moment supplies, masks, were provided. That’s why the mobile clinic was intensified.*
P3


*Everyone is equipped (professionals) with masks and alcohol-based hand sanitizers. We provide masks and bathing kits; we advise, talk, and solve any problem in the clinic. Other problems we partner with other health centers (...) And we try to eliminate the demands (...). The mobile clinic can never end because women are desperate (...) and suffering too much.*
P7


*We provide care with masks and encourage them to wear them too, so they don’t give problems among themselves (...). We orient a lot; we talk about the things happening so that they take better care of themselves.*
P6

### 3.2. Challenges in the Care of Homeless Women during the COVID-19 Pandemic

Despite all the efforts to provide care to homeless people in such a turbulent period as the pandemic, limitations and difficulties were experienced in developing care actions.


*(...) during this pandemic, we managed to get some supplies, but this was not enough. For instance, masks, which they ask for a lot in the streets, but which they also don’t take care of, you provide them, and sometimes you go to the same place and they don’t have them anymore, the same people ask you for more masks, even though they are made of fabric.*
P2


*So, it’s the hand washing orientation because they don’t have many things, “ah, wash your hands all the time,” they don’t even have soap. They don’t have this, you understand? So, we put together a kit with a toothbrush, toothpaste, cream and soap, and a mask, and we gave them that too. And it paid off. It’s over.*
P4

Additionally, structural limitations and a lack of social appliances for the reception of homeless women during the pandemic stand out.


*During the pandemic, there was [a shelter with] 130 vacancies for men, and practically the entire pandemic took place with 13 vacancies for women. So, we see that the public health part of the assistance is the repression that starts at home or even on the streets. So everywhere, women suffer a kind of violence from all sides.*
P1


*There is all the concern to insert [the woman] in Alcohol and Drugs Center for Psychosocial Care; if she likes and accepts it, she can have a place that is a reference for her. She can have adequate food and support. So that would be ideal, but nowadays, with the pandemic, it is very difficult.*
P5

During the pandemic, educational activities with women’s groups were interrupted, interfering with the integral attention to women.

*Because of the pandemic, things are difficult.* [Before the pandemic, I offered a support group for women, but now] *I had a women-only group.*P5

Moreover, exposure of professionals belonging to risk groups was observed, given the need for specific interventions that require experience and qualification considering the complexity of the demands of homeless women.


*(...) We had a pregnant woman that no one could bond with, and I had to intervene even though she couldn’t go out to the street [because she was a risk group]. I went to intervene to take her to prenatal care, but I didn’t have much success because she already had the baby after two weeks. But I think the team is trying to maintain itself and trying to do the best it can.*
P5

Faced with the challenges, professionals use alternative prevention strategies to mitigate the lack of resources.


*(...) some days ago, I already turned to him and said, ‘I’ll make a recipe of how to make a mask (...).’ Put the sleeve of the shirt like this, open a hole here and here, and you put it here. Then he said, ‘Don’t you have something to give us? He said, ‘I do,’ I turned and said, ‘then cut it, fold it in the middle, take it, this size like this, cut a piece this size, open a hole here and another here. You can make your mask by yourself, using an old t-shirt. He looked at me like this and said, ‘you get it.’’*
P4

## 4. Discussion

This study demonstrated the strategies used by the Street Outreach Office team in women’s health care during the COVID-19 pandemic. Our findings showed an intensification of the team’s actions during the COVID-19 pandemic, meeting women’s health needs, delivering hygiene supplies, and using masks and alcohol-based hand sanitizers. Nonetheless, a limitation of the actions was observed regarding the lack of material resources for the continuity of the distribution of supplies, difficulties of the homeless population adhering to preventive measures for not having basic resources, and limitations to the actions of professionals.

The participants’ profile was primarily female, graduated, and with ages ranging from 33 to 63 years old, coinciding with the description of the health teams for homeless people in national and international scope, which also present a composition formed by female members and mostly graduated in the range of 30 to 60 [2,20].

The pandemic exposed social inequalities and highlighted the greater need for attention to the most vulnerable populations, challenging the organization of the health system since the health services were not prepared to face this challenge, evidencing the weaknesses and deficiencies of the public health systems [21]. The increase in the homeless population has highlighted even more the need for public health to address and create strategies to deal with a crucial issue for improving health care, which is the lack of housing [22].

During the pandemic, the Street Outreach Office intensified its actions in the care of the homeless population, including women, with the perspective of assisting them with their health needs and specific demands, in addition to carrying out preventive actions against COVID-19. However, COVID-19 exposed the Brazilian health vulnerabilities. The lack of health investment and absence of federal coordinating actions and policies contributed to the tragically high rates of cases and deaths [23,24]. The inequalities in health work conditions, unprepared workers exposed to high risk of contamination and death and the failure to control the COVID-19 pandemic in the population [23,24]. Insecurity was a recurring feeling for Street Outreach Office teams due to the numerous doubts about how to proceed and implement the protocols developed for the general population with the homeless population since there was none directed explicitly to them. Those in force brought guidelines that were not aligned with the way of life of this population. Remaining in confinement in quarantine periods, maintaining social distancing, and following hygiene protocols (e.g., hand washing) were not measures thought of for those at the margins of society and are rarely remembered or contemplated by previous policies; this was no different for the protocols to combat COVID-19 [25,26]. Thus, the Street Outreach Office’s reorganization to meet the population’s demands stands out in this study.

To provide comprehensive health care, the professionals of the Street Outreach Office team must visit areas where homeless people can usually be found. The professionals must be welcoming and accessible and present in these spaces, striking a balance between the care according to the preconization, rules, and standards of health and the tools and space available for care [20]. Therefore, seeking knowledge about this population facilitates the performance and development of care [20], justifying the need to pay attention to the particularities of homeless women.

In this sense, in the excerpts of the professionals participating in the study, it was possible to identify the concern with prenatal care and monitoring of women during pregnancy because this factor adds one more particularity in the care of women’s health. The higher number of pregnancies among homeless women perceived by professionals can be associated with the barriers to providing contraceptive methods and health actions during the COVID-19 pandemic. The intensified care by the team is justified because the insecurity of the streets, lack of housing, and other factors that leave women in a state of greater vulnerability (e.g., inadequate access to health services, food insecurity, violence, and lack of appropriate places to perform hygiene care among others) represent a greater risk to the mother and child binomial, negatively impacting the health of women and children with consequences that can last a lifetime [27,28,29]. It is considered that the team approach and treatment can be a crucial factor in the adherence of these women to prenatal care [27].

Notably, the teams providing care to the homeless were committed to continuing to provide care despite all the uncertainties of the pandemic. All over the world, professionals were challenged to maintain and expand assistance during the pandemic in order not to leave this population helpless, including the most vulnerable, which are women [11,26]. A survey in the city of Rio de Janeiro, in southeastern Brazil, showed that the Street Outreach Office was responsible for 15.5% of care among respondents during the pandemic, demonstrating that the expansion of actions to this population is urgent [11].

It is important to note that the pandemic added to or maximized barriers that already existed for working with homeless people. In Launceston, Tasmania, the fragmentation of services and lack of professionals to provide free care were indicated as barriers to accessing health care [30]. In Copenhagen, Denmark, the difficulty of working cooperatively with other organizations was also highlighted; however, the professionals who managed to establish a positive network demonstrated their intention to continue and expand this integrated and collaborative way of working [31]. In our study, the Street Outreach Office team highlighted the importance of its work with homeless women, the dialogue with the city’s BHU, and how this service benefits their comprehensive health care because by being a service linked to the Unified Health System, care is provided free-of-charge to the population [14], contributing to minimizing this barrier.

Other studies also mentioned the concern experienced by participants in the care of homeless women with decreased access to health and wellness services, which pointed to the general concern for follow-up care, especially in mental health care. This is related to treating substance dependence and chronic illnesses, shelter availability, and places to provide food, prenatal follow-up groups, and care in general [25,26].

Moreover, our findings also showed that the lack of resources and social inequalities worsened during the pandemic, especially for the street population. While the general population with resources can migrate their activities to the remote and online mode, the activities with the street population cannot happen this way [25]. Given the lack of resources, the deprivation of this population’s access to reliable and trustworthy information is also evident, generating insecurities and difficulty in adhering to protocols correctly [32]. In this sense, the teams assume a vital role of informing while caring, distributing the safety equipment while using them correctly to set an example and foster the incentive to use [32], even in such a precarious situation as it is on the streets. In Denmark, in the first moment of the pandemic, health providers even took on the work of translating the official guidelines so that immigrants who did not speak the language could have access to information [31].

During the pandemic, the challenge of the urgent need to expand the teams of care on the street became evident, either of professionals of the Street Outreach Office or volunteers, as in the cases of care provided by non-governmental organizations [25]. What is more, some health professionals were characterized as a risk group for complications of COVID-19 during the pandemic period, being removed from health work, which further reduced the contingent of professionals in street care [25], especially those who had more experience in working with this population. Conducting this research only with Street Outreach Office teams, without analyzing the homeless women’s perspective, can be considered a limitation of this study. However, the method used allowed the understanding of the challenges in assisting this population, aiming to raise awareness of the professionals and services that make up the health care network.

## 5. Conclusions

The COVID-19 pandemic resulted in the intensification of actions for the health care of the homeless population, including women, with the need to hire new professionals due to the situation’s complexity. Among the problems faced in the care of women, we highlight pregnancy during the pandemic, lack of supplies (e.g., masks, alcohol-based hand sanitizers, and hygiene products) for homeless people, the difficulty of implementing prevention measures given the precarious living conditions on the streets, lack of infrastructure for the care and shelter for women, and the removal of experienced professionals in the at-risk group. Even with all the adversities of facing these problems, the professionals of the mobile health clinic used creative strategies using a handful of resources to prevent COVID-19, contributing, within their limitations, to the care of homeless women.

## Figures and Tables

**Table 1 ijerph-20-01011-t001:** Coding data. Ribeirão Preto—SP, Brazil, 2022.

Initial Codes	Intermediate Codes	Thematic Categories
Meeting between the health workers and homeless women	Street Outreach Office assistance to homeless women	Reorganization of the Street Outreach Office to meet the population’s demands
Recognition of homeless women’s health needs		
Increase of vulnerability for homeless people during pandemic	Need to employ new health workers and expand the Street Outreach Office’s team	
Increase of work demands during pandemic		
COVID-19 prevention measures and health orientation		
Distribution of supplies and inputs for the homeless population		
Limited financial resources to provide health care assistance to homeless population during pandemic	Creation of alternatives to amortize the problem and use of the scarce street resources to protect against COVID-19	Challenges in the care of homeless women during the COVID-19 pandemic
Lack of supplies to offer to the population		
Difficulties of homeless people to take care of their supplies for protection		
Limited shelter’s access to homeless women	Difficult situations to provide the homeless women’s care	
Difficulties to access the health services network		
Barriers to develop educative health activities		
Higher COVID-19 contraction risk and death	COVID-19 burden in homeless women’s health	
Difficulties to provide women’s sexual and reproductive health and rights; unmet need for contraception.		
Pregnancy among homeless women, higher risk of unfavorable maternal and fetal outcomes due the lack of antenatal care during the pandemic.		
Misuse substance		
Violence		

## Data Availability

The data presented in this study are available on request from the corresponding author. The data are not publicly available due to data privacy following the approval by the Research Ethics Committee of the University of São Paulo at Ribeirão Preto College of Nursing, Brazil.

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
