# Peer review of "Assisting Homeless Women in a City in Brazil during the COVID-19 Pandemic in the Context of a Street Outreach Office: The Perceptions of Health Professionals"

_ijerph, 2023, doi:10.3390/ijerph20021011_

Round 1

Reviewer 1 Report

This qualitative paper analyzed interviews with women working as part of a street outreach team providing health care to women experiencing homelessness. There are several points of clarification that it would be helpful to incorporate:

1. It is unclear how the team managed qualitative coding. Did the authors begin with a pre-made codebook, which coders then applied to transcripts? If so, what were the pre-determined codes, and how was fidelity of coding across multiple coders assessed? If coding was more inductive, what was the process for the development of the codebook? More detail here would be helpful for readers to understand whether the analysis was done rigorously.

2. In the introduction, the authors state that homeless people are more vulnerable to COVID-19 because their health is poorer generally, meaning they suffer conditions that put them at high risk of severe illness. But there is also simulation-based and applied research showing that moving into crowded shelters or crowding into small houses with their relatives to avoid sleeping on the street also amplifies their risk of contracting COVID-19. This pathway between homelessness and COVID-19 risk should be discussed further:

https://www.nature.com/articles/s41467-021-22521-5

https://pubmed.ncbi.nlm.nih.gov/34309643/

3. Could the authors clarify what they mean by "floating reading" on page 2? 

4. On line 63, the phrase "since they need to attend complex needs out of office" does not make sense to me. Are the authors trying to say that traveling from place to place providing complex healthcare is tiring for workers?

5. The authors write that, "During the pandemic, educational activities with women’s groups were interrupted, interfering with the integral attention to women." The quote they use to support this is, "Because of the pandemic, things are difficult. I had a women-only group." This quote does not really support their argument - the respondent does not discuss an interruption of the services they provided, or that women were particularly impacted, just that providing the services were difficult and that they worked with women. Please clarify and provide additional supporting quotes.

6. On line 178, did you mean "He looked at me... and said "you CAN help yourself"? or "I can help myself"? It doesn't make much sense as written. 

Author Response

Response to Reviewer 1 comments

Point 1:  a) How the team managed qualitative coding? b) Did the authors begin with a pre-made codebook, which coders then applied to transcripts? If so, what were the pre-determined codes, and how was fidelity of coding across multiple coders assessed? If coding was more inductive, what was the process for the development of the codebook? More detail to understand whether the analysis was done rigorously.

Response 1:  To manage the qualitative coding we used: initial codes and intermediate codes to create the thematic categories. We created a chart  (chart #1) to show the qualitative coding of research.

Point 2: In the introduction, the authors state that homeless people are more vulnerable to COVID-19 because their health is poorer generally, meaning they suffer conditions that put them at high risk of severe illness. But there is also simulation-based and applied research showing that moving into crowded shelters or crowding into small houses with their relatives to avoid sleeping on the street also amplifies their risk of contracting COVID-19. This pathway between homelessness and COVID-19 risk should be discussed further: https://www.nature.com/articles/s41467-021-22521-5 https://pubmed.ncbi.nlm.nih.gov/34309643/

Response 2: We agree with the Reviewer and insert the two references suggested in the first paragraph on the Introduction section.

“Also, thousands of families were exposed to the risk of evictions during the pandemic due the economic crisis and increasing unemployment caused by the restrictive measures. In this sense, there is evidence that moving to crowded shelters or crowding into small houses with family members, to avoid sleeping on the street, were also linked to increasing risk of contracting covid-19. In this way, it appears that evictions are directly and indirectly associated with the risk of contracting covid-19 as well [Leifheit et al. 2021, NANDE et al. 2021]. It is important to mention that evictions were higher among people of color, low income and women;  black poor women had higher rates of eviction when compared to other groups of people [Benfer et al., 2021].”

Point 3: Could the authors clarify what they mean by "floating reading" on page 2?

Response 3: floating reading is part of the data pre-analysis. The researchers carry out skimming reading to start building indicators for the analysis: definition of units, keywords or phrases. According to Minayo (1998), floating reading is the first contact with the text, capturing the content generically. MINAYO, Maria Cecília de Souza. O desafio do conhecimento: pesquisa qualitativa em saúde. 5. ed. São Paulo: Hucitec-Abrasco, 1998.

Point 4: On line 63, the phrase "since they need to attend complex needs out of office" does not make sense to me. Are the authors trying to say that traveling from place to place providing complex healthcare is tiring for workers?

Response 4: in fact, this is a sentence completely disconnected from the main idea of the paragraph. We chose to remove it entirely from the manuscript.

Point 5: The authors write that, "During the pandemic, educational activities with women’s groups were interrupted, interfering with the integral attention to women." The quote they use to support this is, "Because of the pandemic, things are difficult. I had a women-only group." This quote does not really support their argument - the respondent does not discuss an interruption of the services they provided, or that women were particularly impacted, just that providing the services were difficult and that they worked with women. Please clarify and provide additional supporting quotes. 

Response 5: we agree that the way it is presented, the sentence does not make sense.  What we wanted to say was that during the most critical period of the pandemic, the activities that took place in groups were all suspended by the health surveillance of the municipality. The phrase chosen to illustrate this situation meant that

“Because of the pandemic, things are difficult. [Before the pandemic, I offered a support group for women, but now] I had a women-only group. P5”

Point 6: On line 178, did you mean "He looked at me... and said "you CAN help yourself"? or "I can help myself"? It doesn't make much sense as written.

Response 6:  In the original speech, the participant used Brazilian slang to say that the health professional can work even in difficult situations. We substituted the sentence quoted by the reviewer above and used an expression in the English language with the same meaning to translate this speech.

“You can make your mask by yourself, using an old t-shirt. He looked at me like this and said, ‘you get it.’ P4”

Reviewer 2 Report

Dear authors, congratulations on choosing this research topic. Given that you have prepared an exploratory research for publication in a prominent international journal, I believe it is necessary to improve: - Introduction - explain in more detail the importance of carrying out research on the homeless, with a more detailed description of the research carried out so far; - Method - explain in more detail the characteristics of qualitative methodology and the importance of applying the basis of that method in the realization of research on this topic; - results - I consider this the weakest part of the work. Please give more details and explain the results of the research - discussion to connect more clearly its results with previously realized researches; - It is mandatory to state the limitations of the research

Author Response

Response to Reviewer 2 comments

Point 1: Introduction - explain in more detail the importance of carrying out research on the homeless, with a more detailed description of the research carried out so far;

Response 1: We  insert more information in the Introduction section to attend the Reviewer suggestion.

Point 2: - Method - explain in more detail the characteristics of qualitative methodology and the importance of applying the basis of that method in the realization of research on this topic.

Response 2: The authors agree with the reviewer that there were gaps in this information in the description of the manuscript's method and we insert the information:

“The interviews were audio-recorded and transcribed in full, preserving their originality. In the subsequent stage, floating reading was performed, accompanied by successive readings and a thorough exploration of the material to capture the relevant aspects to answer the research objectives. Thematic content analysis was chosen for data analysis [18,19], according to the phases of pre-analysis and data exploration that were thoroughly done and treating the results, inference, and interpretation [18].

Point 3:  results - I consider this the weakest part of the work. Please give more details and explain the results of the research

Response 3: we worked to make the characterization of the participants more robust and better qualify the speeches.

Point 4: discussion to connect more clearly its results with previously realized researches; Response 4:  the discussion was corrected to connect with the results.

Point 5- It is mandatory to state the limitations of the research.

Response 5:  we have inserted a paragraph about the limitations found in this study. Stayed like this:

“Conducting this research only with Street Outreach Office teams, without analyzing the homeless women’s perspective, can be considered a limitation of this study. However, the method used allowed the understanding of the challenges in assisting this population, aiming to raise awareness of the professionals and services that make up the health care network”.

Round 2

Reviewer 2 Report

Thank you. You have been improved the manuscript in line with suggestions.